# Strong interactions between solitons and background light in Brillouin-Kerr microcombs

Menghua Zhang[1,2], Shulin Ding[1,2], Xinxin Li[1,2], Keren Pu[1], Shujian Lei[1], Min Xiao [1] & Xiaoshun Jiang [1] ✉

Dissipative Kerr-soliton combs are laser pulses regularly sustained by a localized solitary wave on top of a continuous-wave background inside a nonlinear resonator. Usually, the intrinsic interactions between the background light and solitons are weak and localized. Here, we demonstrate a strong interaction between the generated soliton comb and the background light in a Brillouin-Kerr microcomb system. This strong interaction enables the generation of a monostable single-soliton microcomb on a silicon chip. Also, new phenomena related to soliton physics including solitons hopping between different states as well as controlling the formations of the soliton states by the pump power, are observed owing to such strong interaction. Utilizing this monostable single-soliton microcomb, we achieve the 100% deterministic turnkey operation successfully without any feedback controls. Importantly, it allows to output turnkey ultra-low-noise microwave signals using a free-running pump.

Dissipative Kerr soliton combs[1–5] in optical resonators have received extensive interests in fundamental research on soliton physics[6–14] as well as applications such as optical coherent communications[15,16], dual-comb spectroscopy[17,18], ultrafast ranging[19,20], optical atomic clocks[21], and optical neural networks[22,23]. In such Kerr-comb systems, the laser pulses are typically sustained by a solitary wave localized on top of a continuous-wave (CW) background[1,4,5]. During that process, the CW background light is provided by an external pump laser, and the loss of Kerr solitons is compensated with the parametric gain produced by nonlinear interactions between the background light and the solitons. Also, in the study of soliton physics, the interactions between CW background light and the solitons usually play important roles in the formations of soliton Cherenkov radiation[6], breathing solitons[8,9], and soliton crystals[10,11]. Additionally, active modulations on the CW background light can be utilized to yield protected solitons[24] or synthesized soliton crystals[25]. However, in previous Kerr soliton combs, the intrinsic interactions between the background light and solitons are typically weak and localized in the spatial overlap region[4,5,26], which means that the formation of solitons has little impact on the amplitude

of the background light. As a result, solitons with different numbers usually share the same CW background field under the same pump condition[5,27–29], which leads to the degeneracy of the soliton existence range for different states, i.e. the multiple stable soliton states exist under the same pump detuning.

To break the degeneracy and deterministically create a single-soliton state with a smooth spectral envelope in an optical microresonator, gradually backward[28] or bidirectional tuning of the injected pump laser frequency[30] is usually required to reduce the soliton number. It has been noticed that spatial mode-interaction also contributes to the single-soliton generation[31]. Recently, spontaneous soliton formation was achieved in an edgeless photonic crystal resonator where the single-soliton state can be directly accessed by breaking up the flat state and bypassing the chaotic state[32]. Although Kerr-soliton microcombs with turnkey operation are more favorable in practice, to date such an operation has only been accomplished by the technique of self-injection locking[33] with a special operating point[34]. Nonetheless, all of those schemes require special optical and electrical controls/feedbacks or

[1]National Laboratory of Solid State Microstructures, College of Engineering and Applied Sciences, School of Physics, Collaborative Innovation Center of Advanced Microstructure, Nanjing University, Nanjing 210093, China. [2]These authors contributed equally: Menghua Zhang, Shulin Ding, Xinxin Li. ✉e-mail: jxs@nju.edu.cn

structures, owing to the presence of multi-stable cavity response within the soliton-generation range.

Unlike previous works, here, we propose and experimentally demonstrate an interesting case with strong interaction between the generated Kerr solitons and the background light in a Brillouin-Kerr microcomb system[13]. This unique physical mechanism can result in rich soliton physics such as effectively breaking the degeneracy of the soliton existence range for different states and hence enabling the generation of a monostable single-soliton microcomb as well as observing hopping between different soliton states. Taking advantage of the monostable and thermally self-stable properties of the Brillouin-Kerr soliton microcomb[13], we further realize a 100% deterministic turnkey single-soliton microcomb without resorting to any complicated feedback controls. Importantly, our scheme for achieving the turnkey single-soliton microcomb not only manifests easy operation but also features a set of intriguing attributes such as ultra-narrow-linewidth comb lines, thermal self-stabilization, low-noise and stable repetition rate, and input pump-power insensitivity. In particular, we

realize a turnkey microcomb-based low-noise microwave signal source at the K-band with a phase noise of −94 dBc/Hz at 1 kHz and −128 dBc/Hz at 10 kHz offset frequency, respectively.

## Results

### Strong interactions between solitons and CW background

In our current Brillouin-Kerr soliton microcomb scheme, the background field is provided by the generated intracavity Brillouin laser rather than the external pump as in the conventional soliton combs. As a consequence, in an ultra-high-quality (ultra-high-Q) microcavity, the generated Kerr-solitons will greatly influence the power of Brillouin laser through optical parametric frequency conversion (Supplementary Section I) and strongly attenuate the intracavity energy of the background light (Fig. 1a). This attenuation of background light subsequently leads to a significant reduction in the parametric gain that is used to sustain the Kerr solitons. Since the strength of this attenuation varies with the soliton number (Fig. 1b), we find that the degeneracy of the existence range between the single soliton state

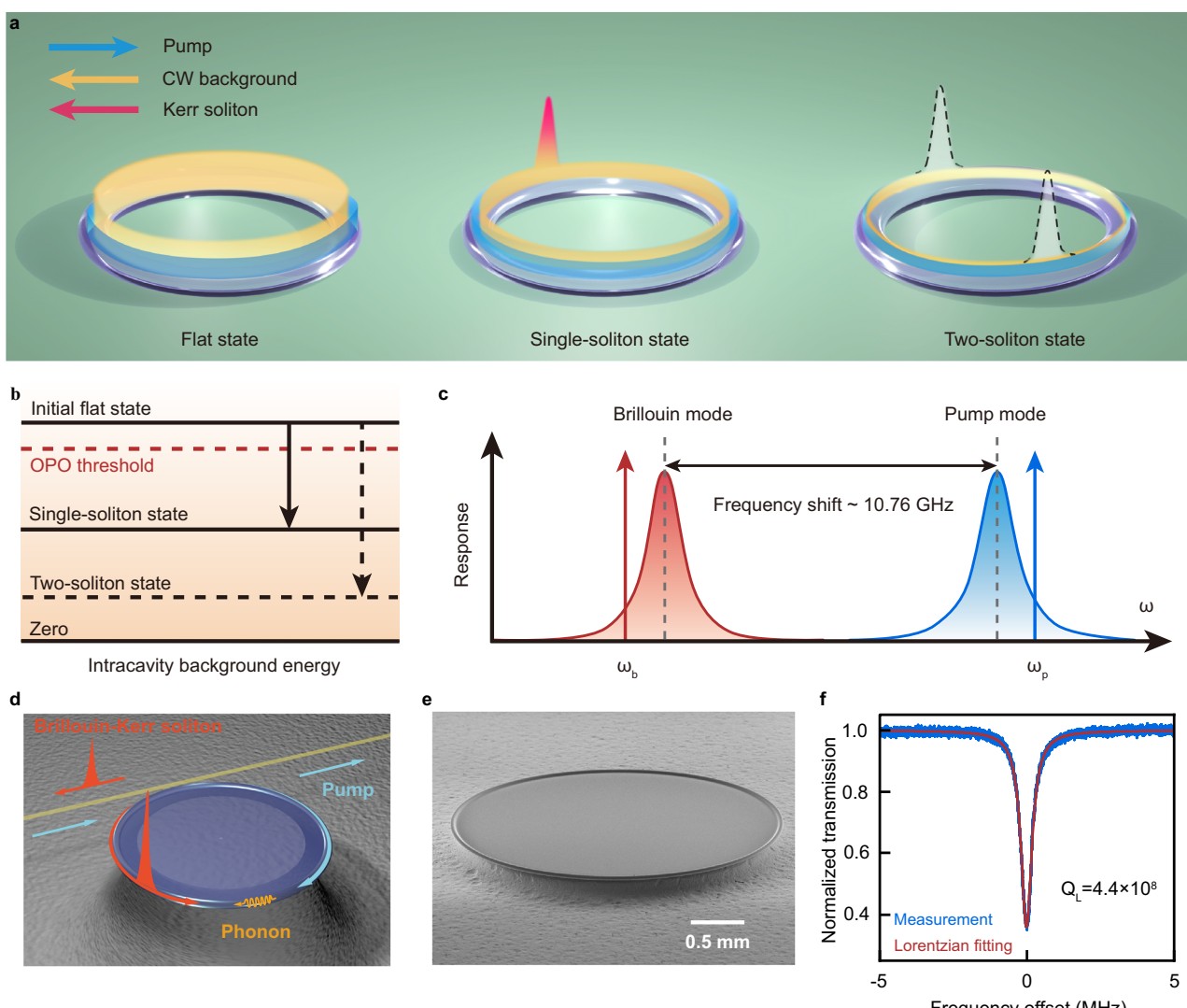

**Fig. 1 | Schematic of monostable single-soliton microcomb generation on the continuous-wave background field provided by the intracavity Brillouin laser in an ultra-high-Q microresonator. a** Illustration of monostable single-soliton formation. Left, the initial continuous-wave (CW) background field before the soliton formation. Middle, single soliton sustained by an attenuated background field. Right, prohibited two-soliton state. **b** Illustration of intracavity-background-field energy levels during the generation of the solitons. OPO, optical parametric oscillation. **c** Illustration of red-detuned Brillouin laser generation via blue-detuned pump. $\omega_p$ and $\omega_b$ stand for the angular frequencies of input pump and Brillouin waves, respectively. **d** Diagram of Brillouin-Kerr soliton generation in a silica microtoroid resonator. **e** The scanning electronic microscope image of the on-chip silica microtoroid used in experiments. **f** Transmission spectrum of the Brillouin mode of interest. $Q_L$, the loaded Q-factor of the Brillouin mode.

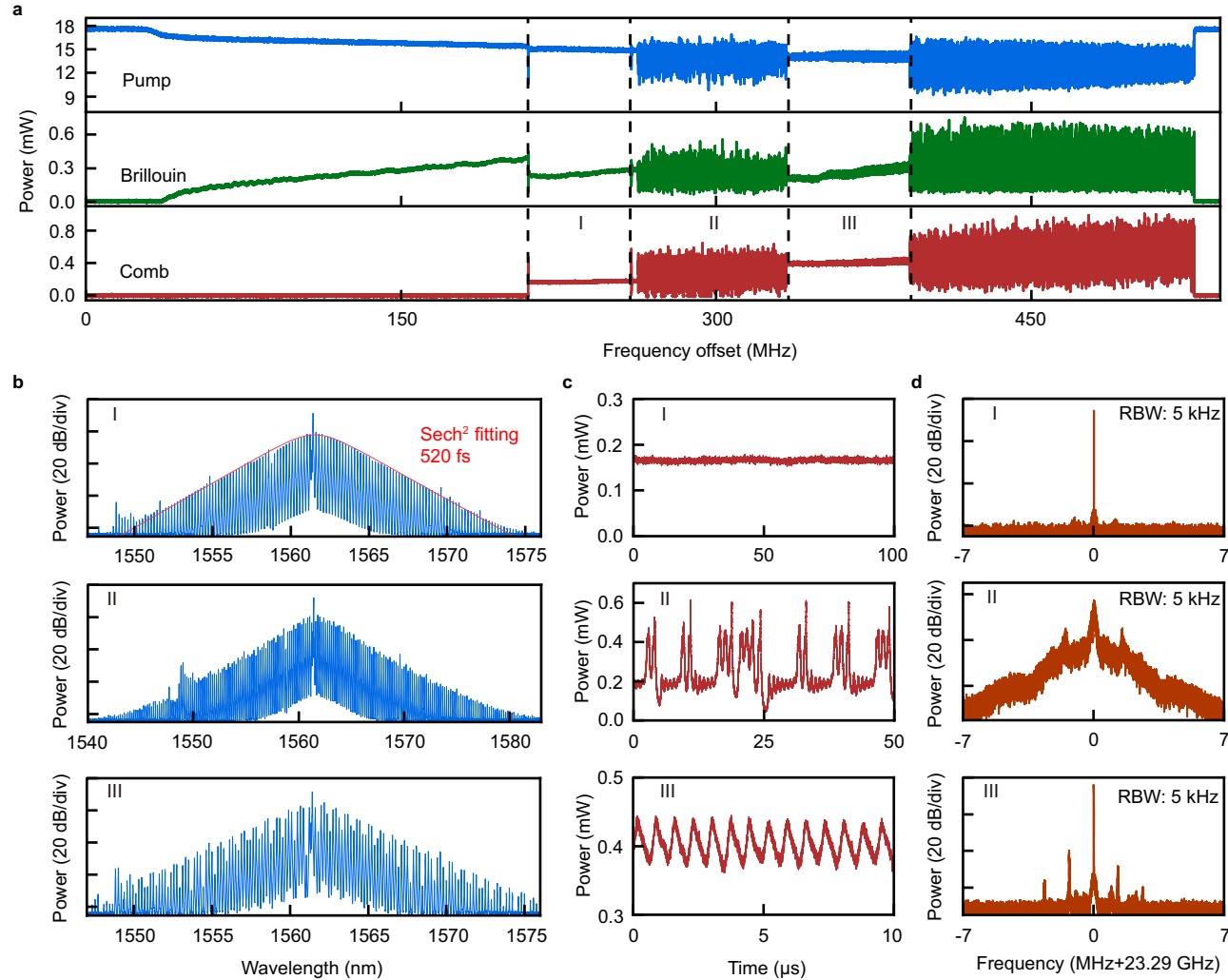

**Fig. 2 | Observation of monostable single-soliton formation in a Brillouin-Kerr microcomb system. a** Typical transmission power spectra of the incident pump (blue), generated Brillouin laser (green), and generated comb (red) in the scanning process by decreasing the pump laser frequency, which sequentially exhibit three distinct transition phases: (I) single-soliton state, (II) intermediate hopping state between single- and two-soliton states, and (III) breathing two-soliton state.

**b** Optical spectra of Phases I-III. **c** Measured temporal output power trace of the generated microcomb versus time in Phases I-III corresponding to (**b**). Dashed line in the middle panel shows the power level of the single-soliton state for comparison. Gray area, the power range of the breathing two-soliton state.
**d** Corresponding radiofrequency (RF) spectra of the microcomb repetition frequency measured with 5-kHz resolution bandwidth (RBW).

and the flat or multi-soliton states will break down. Here, the strong interaction between the generated Kerr solitons and the background light is referred to the situation, in which the intracavity energy of the background light is decreased from above to below the optical-parametric-oscillation (OPO) threshold and becomes low enough to prevent the formation of multi-solitons during the Kerr soliton generation process.

In order to experimentally verify such strong interaction, we employ an ultra-high-Q microtoroid resonator[35,36] with a 2.8-mm-diameter to initiate the Brillouin-Kerr soliton microcomb (Fig. 1d, e), because the higher optical Q-factor will induce a stronger effective Brillouin gain for a fixed pump power[37]. According to the measurements (Fig. 1f), the loaded Q-factor of the Brillouin mode (Fig. 1c, its corresponding mode family is used for soliton microcomb generation) is $4.4 \times 10^8$, while its corresponding intrinsic Q-factor is $5.6 \times 10^8$. In this way, the interaction strength between the Brillouin laser and the generated Kerr-soliton microcomb will become more pronounced (Supplementary Fig. S6), since the generation of Kerr solitons will remarkably reduce the intracavity energy of the Brillouin wave (see detailed discussions in the Supplementary Section I).

## Monostable single soliton
Experimentally, we first input the pump mode with 17.9-mW power and scan the laser frequency. In Fig. 2a, we display the scanned power spectra of the forward transmitted pump, Brillouin laser, and backward comb by decreasing detuning of the laser frequency. As one can see from Fig. 2a, in the course of decreasing the pump frequency, the Brillouin lasing will first appear at a far blue-detuned input pump. Then, the red-detuned intracavity Brillouin laser[13] (Fig. 1c) begins to grow and eventually exceeds the threshold of the OPO (when the output power of the Brillouin laser reaches ~0.39 mW), heralding the emergence of the Kerr microcomb. Here, we can identify the red-detuning of the Brillouin laser by launching a weak probe light while generating the soliton microcombs[38] (details are shown in Supplementary Fig. S13). Surprisingly, at this time, a single-soliton state (Phase I in Fig. 2) immediately emerges from the flat state. To understand the generation of such soliton, we build a model by employing nonlinear coupled-mode equations[13] (Supplementary Section I) with consideration of the processes including stimulated Brillouin scattering, Kerr nonlinearity, and the interaction between them. And, we find that the generated microcomb will affect the intracavity Brillouin laser via parametric frequency conversion and lead to a reduction of the

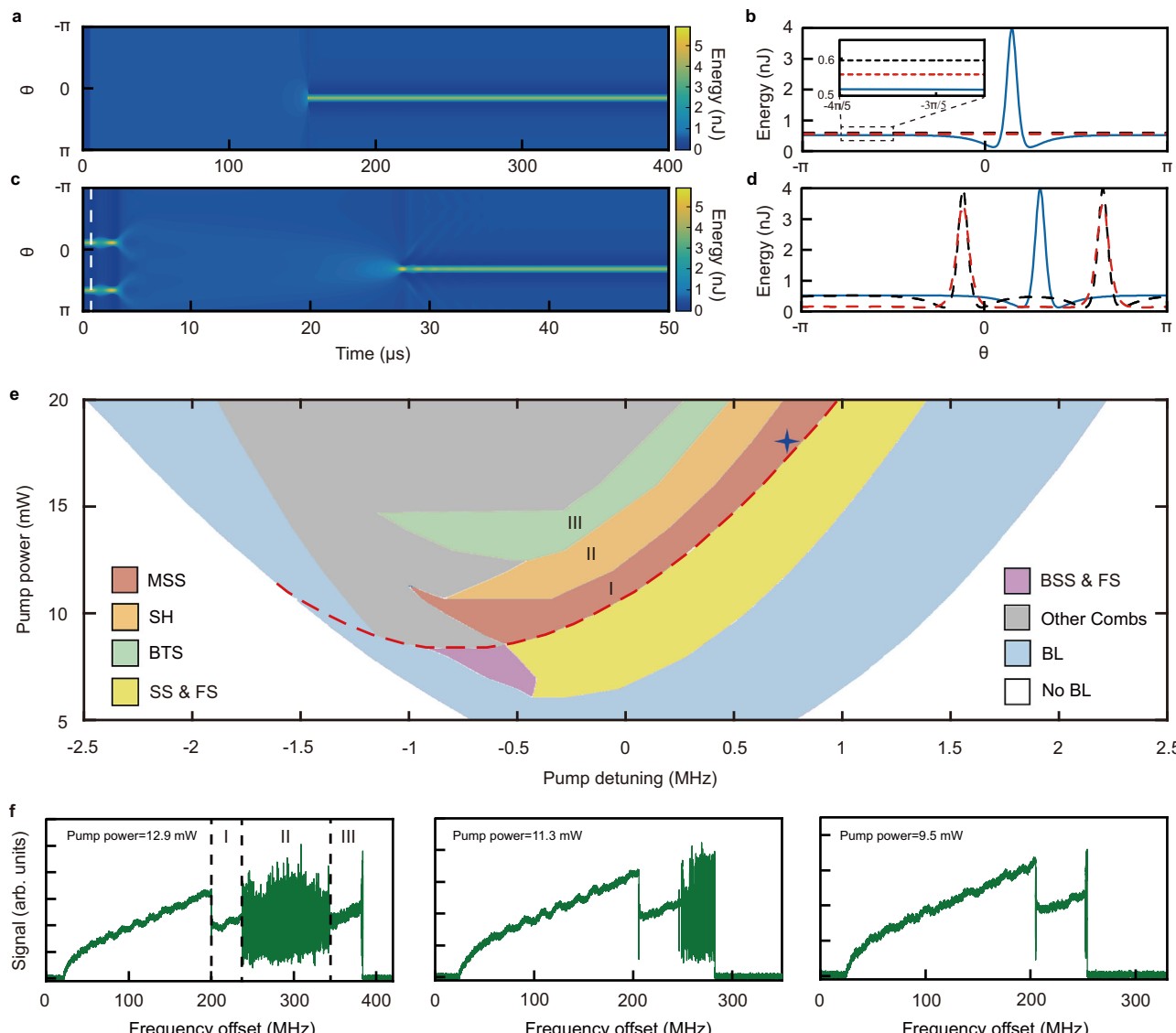

**Fig. 3 | Numerical simulations and transmission power spectra of the generated Brillouin lasers under lower pump powers. a** Monostable single-soliton formation. $\theta$ is the angular coordinate in a co-rotating frame. By repeating the calculations implemented in (**a**), the reliability of such single-soliton generation can be verified. **b** Eventual single-soliton state (blue solid line) formed in (**a**). Black dashed line, intracavity energy of Brillouin laser before the soliton formation. Red dashed line, energy level of the OPO threshold. **c** Two-soliton temporal evolutions with the same parameters of (**a**). **d** Eventual envelope (blue curve) of the reformed single soliton. Black dashed curve, the initial envelope of the solitons. Red dashed curve, the decayed background light and solitons after propagating a short time (at the moment marked by the white dashed line in (**c**)). **e** Theoretical phase diagram displaying the systemic states under different pump detunings and powers. The colored areas indicate different states: red (Phase I), monostable single soliton (MSS); orange (II), soliton hopping (SH) between the single and two solitons; green (III), breathing two solitons (BTS); yellow, coexistence of single-soliton states and flat states (SS & FS); violet, coexistence of breathing single-soliton states and flat states (BSS & FS); gray, other combs; blue, Brillouin laser (BL); white, no BL. The red dashed curve indicates that the generated intracavity Brillouin laser equal to the OPO thresholds. The blue cross highlights the simulated parameters of (**a**)-(**d**) in the phase diagram. **f** Measured transmitted power spectra of the Brillouin lasers under different input pump powers.

intracavity energy of CW background field. In turn, this reduction of background light lowers the parametric gain for the microcomb and further influences the soliton dynamics. Actually, this interaction can be remarkably strong and result in globally reducing the intracavity energy of the background light to be below the OPO threshold (Fig. 1b), which eventually makes the microcomb enter into a stable soliton state (Phase I marked in Fig. 2) directly from the flat state (simulations in Fig. 3a, b and further details can be found in Supplementary Section II). Note that this global damping behavior (Supplementary Section III) of the intracavity energy of the CW background field during the soliton formations cannot occur in a conventional[1,4,29] or even an efficient Kerr soliton microcombs[39] under the fixed pump detuning and power, where the solitons only locally interact with the background light. Additionally, we find that the single-soliton state can be 100% deterministically generated (by repeatedly sweeping the pump laser frequency, see Supplementary Fig. S7), which is quite different from the generation of the conventional Kerr solitons by directly pumping the Brillouin mode (Supplementary Fig. S2). Here, we attributed this behavior to the monostability of the single-soliton state. Since the intracavity energy of the solitons increases with the increase of the soliton number, the amplitude of the intracavity background field becomes monotonically decreased with the increase of the soliton number (Supplementary Section I). Therefore, due to the relatively low initial intracavity Brillouin energy prior to the soliton generation, only the generation of the single soliton can make the microcomb stable and any multi-soliton generation will lead to the intracavity

energy of the background field too low to maintain enough parametric gain for Kerr-soliton propagation (Fig. 1b, and see simulations in Fig. 3c, d and more details in Supplementary Section II). In this case, the single-soliton state is the only stable state in Phase I (the multi-soliton states are prohibited while the flat state is unstable), thence demonstrating the monostability of this single-soliton state. In our experiment, the efficiency of the generated single soliton is around ~1.0% (with ~15% input pump power dropped into the cavity mode), which might be further improved by optimizing the parameters of the system, such as the external coupling rate of pump mode and the mode spacing between pump and Brillouin modes, to increase the input-pump-power absorption[40,41].

### Observation of a soliton hopping state

Further decreasing the pump detuning, we observe the appearance of a novel unstable intermediate state (Phase II in Fig. 2a) between the single-soliton state (Phase I) and the two-soliton breather state (Phase III). In addition, from the middle panel of Fig. 2c, one can see that the output power trace switches between the power levels of single soliton (dashed line) and breathing two solitons (gray area), which indicates that the system hops between these two states. According to our simulation (Supplementary Fig. S5a, b), we find that one additional soliton will form from the CW background light because of the relatively high parametric gain in Phase III of Fig. 2. However, due to the strong interaction between the generated soliton and the background light, the formation of one more soliton will reduce the amplitude of the background field as well as the parametric gain, thus making both solitons unstable and decay. As a consequence, these two solitons will convert back to single soliton again. Since the intracavity energy of the background field with the single-soliton propagation is above the OPO threshold, another soliton will regenerate, and, such, the system starts another cycle. As a result, such an intermediate state also serves as an indicator for separating the single- and two-soliton regions. Alternatively, the observation of this intermediate state implies that only the single-soliton state can exist in Phase I. Finally, all these measured phase transitions displayed in Fig. 2 can been fully confirmed by our theoretical calculations (Supplementary Section IV).

### Simulation of phase diagram

To gain more information on the observed new soliton states, we further simulate the phase diagram (methods see Supplementary Section I, II, IV) of our system. As shown in Fig. 3e, we find that all the observed states (Phases I-III) exist in a wide parametric region, respectively. In addition, one can see that the Phases III, II, and I will sequentially disappear with the decrease of the input pump power. These phenomena can be further confirmed by our experimental observations (Fig. 3f), which demonstrates a simple and convenient way to manipulate the emergence of the soliton states with the pump power. Intriguingly, we notice that the monostable single-soliton state exists when the intracavity Brillouin laser is beyond the OPO threshold (red dashed curve in Fig. 3e). Such new feature is very different from the previous Kerr-soliton combs[28], where the soliton states only exist in the parametric regions below the OPO threshold. In experiments, this property can be shown from the transmission spectra of Figs. 2a and 3f, where the singe-soliton state can be directly generated from the flat state.

### Turnkey operation of the monostable single soliton

Owning to the monostability and thermal self-stability[13] of the Brillouin-Kerr soliton microcombs, our architecture (schematic in Fig. 4a) also enables us to easily implement the turnkey 100% deterministic single-soliton microcomb generation by just linearly increasing the current of an external cavity diode laser (ECDL) from zero up to a certain value. Fig. 4b shows the measured optical transmission spectra of the pump and generated microcomb by increasing the laser

current, in which the soliton steps can be clearly obtained within the pump mode. During the process, both the power and wavelength increase with the enlarged laser current. Moreover, due to the relatively large wavelength changing range, other optical modes (Fig. 4b) can also be observed but have no influence on the microcomb generation. To implement the turnkey operation of the single-soliton microcomb, in the experiment, we linearly increase the laser current and stop the laser at the soliton step (Fig. 4c). In this process, the laser-wavelength increment should be set slightly larger than the resonance redshift induced by the thermo-optic effect[42] when the laser wavelength approaches to the pump mode. This practical arrangement helps to bring the pump light into the cavity mode steadily. In addition, as shown in Fig. 4d, we also confirm that the turnkey process can be repeatably generated by periodically repeating the above process while measuring the comb power and repetition rate. In contrast to the previous methods based on laser self-injection locking and nonlinear resonator response[33], our approach of the turnkey operation becomes insensitive to the input pump power. Experimentally, we have realized the turnkey single-soliton microcomb operation for a pump power ranging from ~9.5 mW to ~19.8 mW, which is limited by the output power of the applied ECDL.

### Low-noise performance of the turnkey soliton microcomb

Benefited from the characteristics of the Brillouin-Kerr soliton microcomb[13], the achieved turnkey single-soliton microcomb here exhibits ultra-narrow linewidth comb lines. As evidenced from Fig. 5a, b, the measured fundamental linewidths of both the Brillouin laser and the comb lines can be as low as the Hertz-level, which is attributed to the ultra-high-Q factors of the employed microtoroid resonator. Compared to previous chip-based Brillouin lasers[40,43,44], our Brillouin laser exhibits relatively broad linewidth, which might be attributed to the phase mismatch of the Brillouin lasing in our system[45].

Besides the above measurements, we further carry out an experiment on generating a low-noise microwave signal in the K-band based on the turnkey single-soliton microcomb after filtering out the reflected pump and Brillouin lasers. The filtered signal is first amplified by an erbium doped fiber amplifier before being sent to a high-speed photodetector for the microwave generation. It is important to highlight that during the measurement of the low-noise microwave signal, the tapered fiber is brought into contact with the microtoroid resonator to enhance system stability. Figure 5c reports the measured single-sideband phase noise of −94 dBc/Hz at 1 kHz, −128 dBc/Hz at 10 kHz and −139 dBc/Hz at 100 kHz offset for a 23.3 GHz carrier. For comparison, Table S1 (Supplementary Section XIII) summarizes the performances of the microcomb-based photonic oscillator. It is evident that our result is significantly lower than that obtained with a turnkey dark pulse[46] and is a record at 10 kHz offset frequency among chip-based platforms[13,46–51]. To characterize the long-term stability of the output turnkey microwave signal, we have also implemented the Allan deviation measurement. As shown in the inset of Fig. 5d, the measured frequency stability for a free-running pump is $1.8 \times 10^{-10}$, $2.0 \times 10^{-10}$, and $6.0 \times 10^{-10}$ over an average time of 0.1 s, 1 s, 10 s, respectively. These performances are fully comparable to the previous result obtained from a chip-based ultra-high-Q microresonator but with the use of Pound-Drever-Hall-locked pump[47].

## Discussion

In conclusion, we have theoretically and experimentally demonstrated a platform with strong interaction between the generated solitons and the CW background light in a dissipative Kerr-soliton microcomb system for the first time. Thanks to this strong interaction, we have demonstrated a monostable single-soliton microcomb on a silicon chip. We have observed a soliton hopping state in which the microcomb hops between the single- and two-soliton

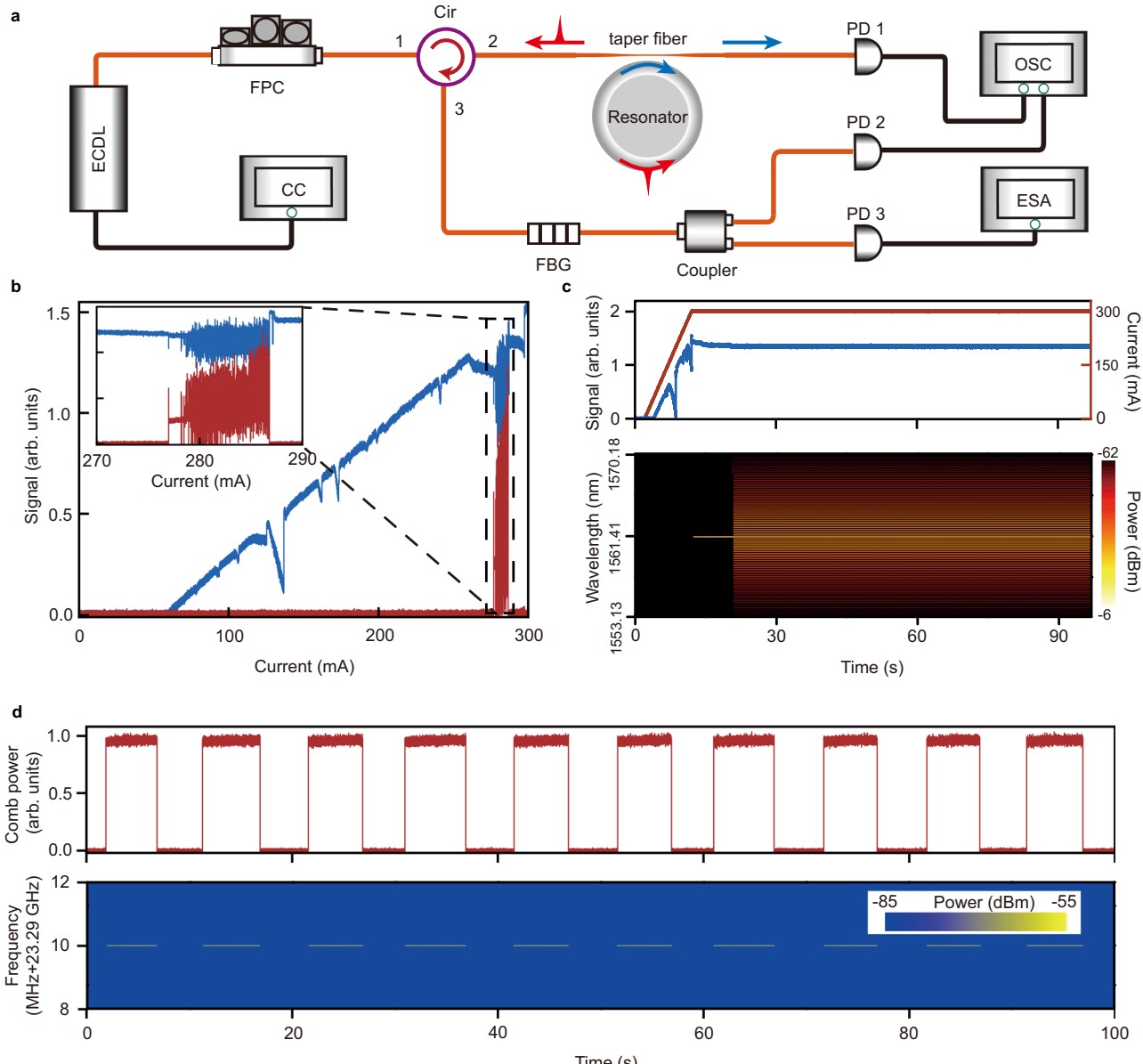

**Fig. 4 | Demonstration of turnkey Brillouin-Kerr single-soliton operation.**
**a** The experimental set-up for implementing the deterministic turnkey operation of the single-soliton microcomb. CC current controller, ECDL external cavity diode laser, FPC fiber polarization controller, Cir optical fiber circular, FBG fiber Bragg grating, PD photodiode, OSC oscilloscope, ESA real-time electric spectrum analyzer. **b** Output pump (blue) and comb (red) powers by merely increasing the driving current of the ECDL. Inset shows zoom view of transmitted pump and comb powers in the region where the microcomb takes place. **c** Real-time trace of transmitted pump power (blue, upper) and current (red, upper). Real-time optical spectra (lower) from current rise until stop to soliton step. **d** Switching on-off comb power (upper) via trapezoidal-wave modulation and real-time RF beatnote signal (lower), which is further monitored by ESA. The RBW is 5 kHz.

states. In contrast to the conventional Kerr solitons, the observed soliton states can exist with the intracavity energy of the generated Brillouin laser beyond the OPO threshold. Moreover, we have demonstrated that the appearance of the soliton states can be controlled by the input pump power. With the monostable Kerr-soliton platform, we have further realized a 100% deterministic turnkey single-soliton microcomb on a silicon chip with superior performance as well as a soliton-microcomb-based turnkey ultra-low-noise microwave signal source. In the future, with the developments of integrated narrow linewidth lasers[52] and waveguide-integrated high-Q microresonators[53], a fully integrated turnkey deterministic single-soliton system based on the monostable single-soliton microcomb is expected to be more desirable in practical applications. In addition, our scheme also provides a versatile platform for studying novel soliton physics with strong interactions between the solitons and the background light.

## Methods

### Silica microtoroid resonator fabrication and coupling
The chip-based microtoroid resonator is fabricated by use a $CO_2$ laser to reflow the edge of a silica disk with a diameter of 3 mm[35,36]. To avoid the stress induced buckling of the microdisk during the $XeF_2$ undercut, we use a 12-μm-thick oxide layer for the fabrication[36]. The measured second-order dispersion of the Brillouin mode family is approximately $2\pi \times 23.7$ kHz (See Supplementary Fig. S11). Experimentally, we use a tapered fiber to evanescently couple light into the optical microresonator[54]. The coupling rates of the pump mode and Brillouin mode can be optimized by adjusting

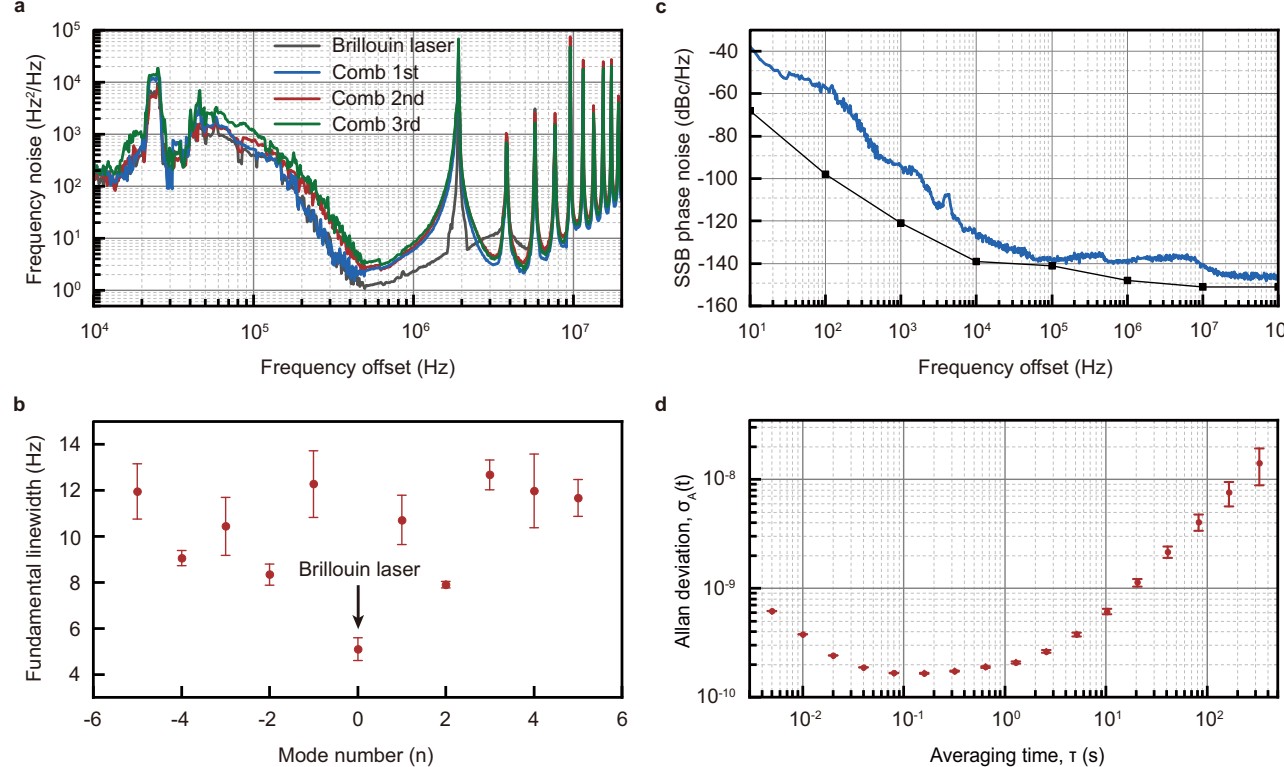

**Fig. 5 | Low-noise performance of the turnkey single-soliton microcomb.**
**a** Single-sided frequency noises of the Brillouin laser and the comb lines adjacent to the Brillouin laser. **b** Measured fundamental linewidths of 10 comb lines and the Brillouin laser. The error bars indicate standard deviation. **c** Single-sideband (SSB)

phase noises of the soliton repetition frequency. The measurement floor of the phase noise analyzer is plotted as black line connecting square dots. **d** The measured Allan deviation of the soliton repetition rate. The error bars indicate standard deviation.

the diameter of the taper fiber and its position relative to the microtoroid resonators.

## Characterization of the soliton-microcomb-based turnkey ultra-low-noise microwave signal

To generate the low noise microwave signal, the generated soliton microcomb is sent to a fiber Bragg grating (0.2 nm bandwidth) to filter out the reflected pump laser and the Brillouin laser. The filtered soliton microcomb is amplified by an erbium-doped fiber amplifier (Amonics, AEDFA-23) and then photomixed by high-speed photodetector (Finisar, XPDV2120RA) with a bandwidth of 40 GHz. The generated microwave signal is measured using a commercial phase noise analyzer (AnaPico, APPH20G). Also, the frequency deviation of the generated microwave signal is measured using a frequency counter (Tektronix, MCA3027).

## Data availability
The data that support the plots within this paper are available on Figshare (https://doi.org/10.6084/m9.figshare.25140752). All other data used in this study are available from the corresponding author upon request.

## Code availability
The simulation and computational codes of this study are available from the corresponding author upon request.

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

## Acknowledgements

This research was supported by the National Key R&D Program of China (2021YFA1400803); National Natural Science Foundation of China (NSFC) (61922040, 12104224, 12341403); Guangdong Major Project of Basic and Applied Basic Research (2020B0301030009); Zhangjiang Laboratory; Natural Science Foundation of Jiangsu Province (BK20221440).

## Author contributions

X.J. conceived the idea and supervised the experiment with M.X. M.Z., X.L. and K.P. implemented the experiment under the guidance of X.J. and M.X. S.L. fabricated the devices. S.D. performed the theoretical model. All authors participated in the discussion of the project and analysis of experimental data. X.J., S.D. and M.Z. wrote the manuscript with contributions from all authors.

## Competing interests

The authors declare no competing interests.
