## [Peer Review File · Nature Communications]

Strong interactions between solitons and background light in Brillouin-Kerr microcombsReviewer #1 (Remarks to the Author):

Review of 'Strong interactions between solitons and background light in Kerr microcombs'
by Menghua Zhang, Shulin Ding, Xinxin Li, Keren Pu, Shujian Lei, Min Xiao, and Xiaoshun Jiang.

The paper demonstrates an elegant new method to directly generate single solitons in a silica micro-resonator without exciting other states, as it is the norm with conventional pumping. The authors achieve 100% deterministic operation without the need of feedback controls. Since the authors generate first a low-noise Brillouin laser as an intermediate step, this enables exceptional noise characteristics for the generated single soliton. In addition, the authors show turnkey generation of ultra-low-noise microwave signals using the reported approach. The theoretical model is sound and reproduces very well the main experimental findings of the paper, in particular the reported new route for the excitation of single soliton states. I believe this work constitutes an important step forward in the deterministic generation of single-solitons in micro-resonators. Despite the great quality of the paper, several points should be addressed before I can recommend it for publication in Nature Communications. As a general remark, I would suggest the authors use less qualitative statements and replace them by quantitative explanations with the help of their model.

1. Perhaps the main claim of the paper is that in the reported approach the interaction between the background light and the solitons is strong and enables the direct generation of single solitons. The authors say that this is in sharp contrast with the usual approach for soliton generation where the interaction between the background light and the solitons is weak and localized hence the formation of solitons has little impact on the amplitude of the background light. How would the authors quantify what is a strong and a weak interaction?. Is this related to the depletion of the Brillouin laser?.
2. From Figure 2(a), the pump power seems to decrease by 35-40%. How is this value associated with the efficiency of the Brillouin and comb generation process?.
3. Brillouin lasers have been generated in micro-resonators with an efficiency surpassing 90% [1]. How does this paper compare to the previously obtained linewidth of the Brillouin laser in [1-3]?. Is there an optimum conversion efficiency value (Brillouin laser power/input pump power) that one should target to get single solitons?. The authors state that the deterministic single-soliton generation is insensitive to the input pump power: could they clarify this?
4. Again in Figure 2(a): it is not possible to provide absolute values instead of arbitrary units?. Figure S2 shows a particular case with actual power values in mW. From this example the efficiency is around 10%, is that correct?. How representative is this case?.
5. From the paper: 'Note that, before the soliton generation, the calculated intracavity energy of the CW background field is as high as ~ 0.51 nJ (higher than the OPO threshold ~ 0.42 nJ), however, after the formation of the single soliton, the intracavity energy of the background field is attenuated to as low as ~ 0.29 nJ (lower than the OPO threshold), which numerically confirms the strong interaction between the generated soliton microcomb and the background light.'
To the best of my knowledge, the strongest damping of the background light experimentally obtained is 55% [4]. How does this paper compares to [4]?
6. Since the authors do not mention any cascaded Brillouin generation, I assume that this is due to the fact that only the first Brillouin order is resonant and higher order waves do not match any resonance?. The authors should clarify that in the paper.
7. If there is no higher order Brillouin generation, that would mean that the linewidth of the Brillouin laser will scale with the input pump power?. Can the authors clarify this?
8. How the noise characteristics of the generated microwave signal compare with other reports using

combs generated in micro-resonators [5-7].

9. In Figure 4, what is the bandwidth of the FBG?

10. The chaotic states in Figure S1 refer to Modulation Instability states, spatio-temporal chaos?. Is it possible to generate soliton crystals or Turing rolls in the architecture proposed in this paper?

11. The reference list lacks important papers related to the claims reported in this contribution. They should include those references: [1-7].

1. J. Li, H. Lee, T. Chen, and K. J. Vahala, "Characterization of a high coherence, Brillouin microcavity laser on silicon," *Optics Express* 20, 20170-20180 (2012).
2. Gundavarapu, S., Brodnik, G.M., Puckett, M. et al. Sub-hertz fundamental linewidth photonic integrated Brillouin laser. *Nature Photon* 13, 60–67 (2019).
3. Wang, H., Wu, L., Yuan, Z., & Vahala, K. (2021, May). Towards milli-Hertz laser frequency noise on a chip. In *CLEO: Science and Innovations* (pp. SF2O-2). Optical Society of America.
4. Boggio, J.M.C., Bodenmüller, D., Ahmed, S. et al. Efficient Kerr soliton comb generation in micro-resonator with interferometric back-coupling. *Nat Commun* 13, 1292 (2022).
5. Liu, J., Lucas, E., Raja, A.S. et al. Photonic microwave generation in the X- and K-band using integrated soliton microcombs. *Nat. Photonics* 14, 486–491 (2020).
6. Kwon, D., Jeong, D., Jeon, I. et al. Ultrastable microwave and soliton-pulse generation from fibre-photonic-stabilized microcombs. *Nat Commun* 13, 381 (2022).
7. Kuse, N., Nishimoto, K., Tokizane, Y. et al. Low phase noise THz generation from a fiber-referenced Kerr microresonator soliton comb. *Commun Phys* 5, 312 (2022).

Reviewer #2 (Remarks to the Author):

The authors describe the unique dynamics of the Brillouin Kerr microcomb generation experimentally and numerically, whose first demonstration is presented in their previous work (Y. Bai, et al., *PRL* 126, 063901 (2021)). The authors discuss the transition processes of the comb and the physics behind them from the viewpoints such as the gain competition and phase diagram. One of the critical features of the comb is the reliable (consistently reproducible) turnkey initiation of a single soliton state by simply adjusting the pump laser current, which is experimentally confirmed. In addition, the authors perform the microwave generation at K-band (23 GHz), and low levels of phase noise and frequency instability are achieved thanks to the laser noise suppression via the Brillouin scattering and the self-thermal stability of the comb, whose results are consistent with the demonstration in their previous study.

Although the described physics is interesting and the comb's nature is intriguing for applications, I think the current edition of the work is not significant enough to be published in *Nature Communications* mainly due to two reasons: The comb and its effects, including low noise microwave generation, have been reported in their published work, and the described explanation about the physics in this study lacks some clarification. The followings are specific concerns, questions, and suggestions.

-The red-detuning of the Brillouin light can be a more critical factor for directly accessing the soliton states than the interaction between background wave and combs. Discussion about blue-detuned Brillouin pumping should be provided to deepen the understanding of physics rather than keeping the red-detuned condition untouched prerequisite. Note that the intensity of CW background light is not fixed even in conventional numerical frameworks (as mentioned in supplementary material around line 113), where the external power flow is usually taken into account via detuning (phase) terms. The interaction between background wave and comb exists, not depending on if you supply driving power directly or indirectly.

-The use of the word "degeneracy" in "the degeneracy of different soliton states" is confusing because each state has a different energy level, as the authors mention in the article (i.e., multi-stability and degeneracy should be different). Note that I agree that selectively accessing soliton states is difficult in standard microcomb generation, where we need to go through chaotic states.

-Can you comment on the effects and requirements of resonator dispersion?

-Can you indicate what power level in Fig. 3e corresponds to the experimental condition for each trace of Fig. 3f, approximately?

-Do you experimentally observe breathing single soliton states (SS) obtained in your numerical simulation (Fig. 3e), though the corresponding state transition is not evident in the time evolution of transmittance?

Minor ones

-Please add an explanation how you couple light into the resonator (e.g., tapered fiber, integrated waveguide).

-What is the origin of bumps on frequency and phase noise curves at around 2 MHz offset? Can you show the sensitivity limit of the measurement?

Reviewer #3 (Remarks to the Author):

This manuscript, "Strong Interaction between Solitons and Background Light in Brillouin-Kerr Microcombs," presents both experimental and numerical investigations of novel phenomena in Kerr soliton microcombs, specifically within the unique Brillouin-Kerr microcomb system. Unlike conventional microcomb setups, the driving source for soliton microcombs in this system relies on intracavity Brillouin light propagating in the backward direction, which offers several advantages over an externally provided forward pump. Although I acknowledge that the results presented are valuable and relevant to the microcomb research community, I have some concerns regarding the manuscript's overall scientific impact, particularly in the context of Nature Communications. I have listed specific questions and comments below.

1) From what I understand, the deterministic emergence of a single-soliton state from a flat state (only Brillouin background) is a key result of this paper. However, I believe that the underlying principle is not sufficiently explained in the current manuscript. Given that intracavity energy is consumed and subsequently reduced to form solitons through OPO processes, it is only natural for the reduction of the background field (i.e., pump power) to affect the potential number of solitons. This interaction does not seem to be a unique feature of the Brillouin-Kerr system, and, unfortunately, I cannot identify significant novelty in this finding.

2) In relation to the previous comment, do multi-soliton states initially emerge instead of a single-soliton state when a stronger pump power is provided (e.g., >100 mW), as reported in the authors' previous paper (PRL, 126, 063901 (2021))?

3) On page 5 (paragraph 5), the authors describe that higher Q-factors lead to a more pronounced effective Brillouin nonlinearity, thereby enhancing the interaction strength. What is meant by "effective Brillouin nonlinearity" in this context? Although high-Q factors do lower the threshold power of Brillouin lasing, it is unclear how this effect directly relates to the interaction strength between Kerr combs and the Brillouin background field.

4) It would be beneficial to include a comparison table of phase-noise performances from previous studies (particularly in the same material platforms).

Minor suggestions:

1) I recommend adding the specific term "Brillouin-Kerr" to the title.

2) The Methods section could be expanded upon. For example, the fabrication methods of resonators and experimental details of soliton generation and measurements could be described in greater detail.

Response to Reviewer 1:

In Reviewer 1's Report: The paper demonstrates an elegant new method to directly generate single solitons in a silica micro-resonator without exciting other states, as it is the norm with conventional pumping. The authors achieve 100% deterministic operation without the need of feedback controls. Since the authors generate first a low-noise Brillouin laser as an intermediate step, this enables exceptional noise characteristics for the generated single soliton. In addition, the authors show turnkey generation of ultra-low-noise microwave signals using the reported approach. The theoretical model is sound and reproduces very well the main experimental findings of the paper, in particular the reported new route for the excitation of single soliton states. I believe this work constitutes an important step forward in the deterministic generation of single-solitons in micro-resonators.

Our Response: We thank the reviewer for the favorable comments on our work.

In Reviewer 1's Report: Despite the great quality of the paper, several points should be addressed before I can recommend it for publication in Nature Communications. As a general remark, I would suggest the authors use less qualitative statements and replace them by quantitative explanations with the help of their model.

Our Response: We thank the reviewer for this very helpful suggestion. In the revised manuscript, we have followed the reviewer's suggestions and used quantitative statements to explain our work such as quantifying the strong interaction between the background light and generated Kerr solitons, showing the conversion efficiency from pump to Brillouin laser and comb powers, and providing absolute values of transmitted power instead of arbitrary units in our experimental diagrams.

In Reviewer 1's Report: Perhaps the main claim of the paper is that in the reported approach the interaction between the background light and the solitons is strong and enables the direct generation of single solitons. The authors say that this is in sharp contrast with the usual approach for soliton generation where the interaction between the background light and the solitons is weak and localized hence the formation of solitons has little impact on the amplitude of the background light. How would the authors quantify what is a strong and a weak interaction? Is this related to the depletion of the Brillouin laser?

Our Response: We thank the reviewer for bringing up this important issue. In our system, the strong interaction is referred to the situation that the intracavity energy of the background light reduces from above the optical-parametric-oscillation (OPO) threshold to below during the Kerr-soliton generations. And, this is surely related to the depletion of the Brillouin laser. To further clarify the strong interaction between the generated Kerr solitons and background field in our system, we have modified the sentence: "As a consequence, in an ultra-high-quality (ultra-high-Q) microcavity, ... thereby indicating a strong interaction between the background field and generated Kerr solitons." to "As a consequence, in an ultra-high-quality (ultra-high-Q) microcavity, the generated Kerr-solitons will greatly influence the power of Brillouin laser through optical parametric frequency conversion (Supplementary Section I) and strongly attenuate the intracavity

energy of the background light (Fig. 1a). This attenuation of background light subsequently leads to a significant reduction in the parametric gain that is used to sustain the Kerr solitons. Here, we refer the strong interaction between the generated Kerr solitons and the background light to the situation that the intracavity energy of the background light is decreased from above the optical-parametric-oscillation (OPO) threshold to below it during the Kerr soliton generation process.” in page 4 of the revised manuscript.

In Reviewer 1's Report: From Figure 2(a), the pump power seems to decrease by 35-40%. How is this value associated with the efficiency of the Brillouin and comb generation process?

Our Response: We thank the reviewer for bringing up this important question. We think the efficiency of the Brillouin and comb should be improved with the increase of the dropped pump power if other parameters of the system are fixed, since only this part of the pump is used for the conversion of the Brillouin laser and Kerr combs. Actually, the decrement of the pump corresponds to the pump detuning for single-soliton generation, which depends on the parameters of our system such as the external coupling rate and the mode spacing between pump and Brillouin modes. In our new measurement, the measured efficiency of the single soliton is ~ 1.0% with 85% transmission (or 15% decrement) of the input pump (Fig. 2a). To address this comment, we have added a sentence: “In our experiment, the efficiency of the generated single soliton is around ~1.0% (with ~ 15% input pump power dropped into the cavity mode), which might be further improved by optimizing the parameters of the system, such as the external coupling rate of pump mode and the mode spacing between pump and Brillouin modes, to increase the input-pump-power absorption^{39,40}.” in page 7 of the revised manuscript.

In Reviewer 1's Report: Brillouin lasers have been generated in micro-resonators with an efficiency surpassing 90% [1]. How does this paper compare to the previously obtained linewidth of the Brillouin laser in [1-3]? Is there an optimum conversion efficiency value (Brillouin laser power/input pump power) that one should target to get single solitons? The authors state that the deterministic single-soliton generation is insensitive to the input pump power: could they clarify this?

Our Response: We thank the reviewer for raising these important questions. Here, we response them one by one as follows:

1. How does this paper compare to the previously obtained linewidth of the Brillouin laser in [1-3]?

We are grateful to the reviewer for pointing out these references to us. In comparison with these references, our Brillouin laser exhibits relatively broad linewidth. We attributed this to the phase mismatch (*Optica* **7**, 1150-1153 (2020)) of the Brillouin lasing in our system. To address this comment, we have added a sentence: “Compared to previous chip-based Brillouin lasers^{39,42,43}, our Brillouin laser exhibits relatively broad linewidth, which might be attributed to the phase mismatch of the Brillouin lasing in our system⁴⁴.” in page 10 of the revised manuscript. We have also added four references (including the three references pointed out by the reviewer and another one) in the revised main text.

2. Is there an optimum conversion efficiency value (Brillouin laser power/input pump power) that one should target to get single solitons?

We thank the reviewer for bringing up this very interesting question. In our experiment, the conversion efficiency of the Brillouin laser is only ~2.2% (about 0.39 mW, see Fig. 2a). To achieve the single soliton, the generated Brillouin laser should exceed the OPO threshold even with a relative low conversion efficiency. Experimentally, we find that the single soliton can be obtained when the output power of the Brillouin laser is ~0.39 mW, which indicates the intracavity energy of Brillouin laser exceeds the threshold of the OPO. To address this comment, we have modified the sentence: “Then, the red-detuned intracavity Brillouin laser¹³ (Fig. 1c) begins to grow and eventually exceeds the threshold of the OPO, heralding the emergence of the Kerr microcomb.” to “Then, the red-detuned intracavity Brillouin laser¹³ (Fig. 1c) begins to grow and eventually exceeds the threshold of the OPO (when the output power of the Brillouin laser reaches ~0.39 mW), heralding the emergence of the Kerr microcomb. Here, we can identify the red-detuning of the Brillouin laser by launching a weak probe light while generating the soliton microcombs³⁷ (details are shown in Supplementary Fig. S10).” in the page 6 of the revised manuscript.

3. The authors state that the deterministic single-soliton generation is insensitive to the input pump power: could they clarify this?

We thank the reviewer for bringing up this question. The “insensitive to the input pump power” refers to the turnkey single-soliton microcomb than can be obtained in a wide range pump power (9.5 mW to 19.8 mW). However, in the previous work based on the laser self-injection locking (*Nature* **582**, 365 (2020)), the turnkey operation of the sing-soliton microcomb can be only achieved at a special operating point. To make our statement more clear, we have modified the sentence: “In contrast to the previous methods based on laser self-injection locking and nonlinear resonator response³², our approach becomes insensitive to the input pump power.” to “In contrast to the previous methods based on laser self-injection locking and nonlinear resonator response³², our approach of the turnkey operation becomes insensitive to the input pump power.” in the page 10 of the revised manuscript.

In Reviewer 1’s Report: Again in Figure 2(a): it is not possible to provide absolute values instead of arbitrary units? Figure S2 shows a particular case with actual power values in mW. From this example the efficiency is around 10%, is that correct? How representative is this case?

Our Response: We thank the reviewer for this helpful comment. We have followed the reviewer’s suggestion and replaced the arbitrary units in Fig. 2a, c with absolute values in the revised manuscript. We also sincerely thank the reviewer for carefully reading our Supplementary Information. Following the reviewer’s advice, we have carefully compared our numerically simulated conversion efficiency with the experimental result. In the original manuscript, the measured pump-to-comb conversion efficiency was ~2% while the simulated value was ~5% in the original Fig. S2. This deviation was mainly originated from the improperly chosen of the simulation parameters such as the overlap factor of the pump and Brillouin modes as well as the cavity mode volumes. In the revised Supplementary Information, we have optimized the parameters for the numerical simulations. And now, the simulated result is in well agreement with the new experimental result (both are 1-2%).

In Reviewer 1's Report: From the paper: 'Note that, before the soliton generation, the calculated intracavity energy of the CW background field is as high as ~ 0.51 nJ (higher than the OPO threshold ~ 0.42 nJ), however, after the formation of the single soliton, the intracavity energy of the background field is attenuated to as low as ~ 0.29 nJ (lower than the OPO threshold), which numerically confirms the strong interaction between the generated soliton microcomb and the background light.' To the best of my knowledge, the strongest damping of the background light experimentally obtained is 55% [4]. How does this paper compares to [4]?

Our Response: We thank the reviewer for pointing out the Ref. [4] (*Nat. Commun.* **13**, 1292 (2022)) to us. In that reference, the authors demonstrated an efficient Kerr soliton microcomb with a 55% pump-to-comb conversion efficiency. However, during that process, the global damping of the amplitude of CW background light was not observed (this can be learned from Fig. S8b, d in the Supplementary Information of Ref. [4]), since the solitons only locally influenced the CW background. To clarify the difference between the damping of the input pump light and the damping of CW background field, we have added a sentence: "Note that this global damping behavior of the intracavity energy of the CW background field during the soliton formations cannot occur in conventional^{1,4,28} or even efficient Kerr soliton microcombs³⁸ with fixed pump detuning and power, where the solitons only locally interact with the background light." in page 6 of the revised manuscript.

In Reviewer 1's Report: Since the authors do not mention any cascaded Brillouin generation, I assume that this is due to the fact that only the first Brillouin order is resonant and higher order waves do not match any resonance? The authors should clarify that in the paper.

Our Response: We thanks the reviewer for the very constructive comments and his/her deep understanding of our work. To clarify this issue, we have followed the reviewer's suggestion and added a sentence: "From Fig. S6, we can deduce that the single-soliton state is generated via the first-order Brillouin laser and there is no cascaded Brillouin laser generated." as well two measured optical spectra as Fig. S6 in page 11 of the revised Supplementary Information.

In Reviewer 1's Report: If there is no higher order Brillouin generation, that would mean that the linewidth of the Brillouin laser will scale with the input pump power? Can the authors clarify this?

Our Response: We thank the reviewer for bringing up this very intriguing comment. We have followed the suggestion of the reviewer and measured the linewidth of the Brillouin laser with different pump power as shown in Fig. S7. Interestingly, we found that the linewidth of the laser nearly the same with different pump powers. This phenomenon may be attributed to the power clamping (as shown in Fig. S5 of the revised Supplementary Information) of the Brillouin laser due to the generation of the soliton. To address this important issue, we have added the measured results as Fig. S7 in the Supplementary Information and added a new section named "Fundamental linewidth measurement of the Brillouin laser" in page 11 of the revised Supplementary Information.

In Reviewer 1's Report: How the noise characteristics of the generated microwave signal compare with other reports using combs generated in micro-resonators [5-7].

Our Response: We thank the reviewer for bringing up this helpful question. We are also aware of a similar comment from Reviewer 3. To address this question, we have added a table in the Supplementary Information to compare the phase noise of the generated microwave signal to other reports. As shown in the table, we can find that our result is a record phase noise at 10 kHz offset frequency for all on-chip microcomb-based low-noise microwave signal and only larger than that obtained in the MgF₂ crystalline resonator with complex locking technique. We have also added several sentences: “For comparison, Table S1 (Supplementary Section XI) summarizes the performances of the microcomb-based photonic oscillator. It is evident that our result is significantly lower than that obtained with a turnkey dark pulse⁴⁵ and is a record at 10 kHz offset frequency among chip-based platforms^{13,45-50}.” in page 11 of the revised manuscript.

In Reviewer 1’s Report: In Figure 4, what is the bandwidth of the FBG?

Our Response: We thank the reviewer for bringing up this question. In our experiments, we use 0.2 nm bandwidth fiber Bragg grating (FBG) to filter out the Brillouin laser for microwave generation. In the revised manuscript, we have added a sentence: “To generate the low noise microwave signal, the generated soliton microcomb is sent to a fiber Bragg grating (0.2nm bandwidth) to filter out the reflected pump laser and the Brillouin laser.” in the “Methods” section of the revised manuscript.

In Reviewer 1’s Report: The chaotic states in Figure S1 refer to Modulation Instability states, spatio-temporal chaos? Is it possible to generate soliton crystals or Turing rolls in the architecture proposed in this paper?

Our Response: We thank the reviewer for bringing up this insightful question. According to our numerical simulations (see Figs. S3e, f), we found that the chaotic states in Fig. S1 are different from the typical modulation instability states or spatio-temporal chaos (*Phys. Rev. X* **9**, 011054 (2019)), since the observations of chaotic temporal oscillations of the background field and stochastic switching of soliton numbers. Such chaotic motions may be attributed to the interaction between the solitons and background field. To address this comment, we have added some discussions “Intriguingly, one may find that the chaotic states (corresponding to Phase VI in Fig. S1) shown in Fig. S3e, f behave different from the typical modulation instability states or spatio-temporal chaos⁸, because of observing of chaotic temporal oscillations of the background field and stochastic switching of soliton numbers (see Fig. S3e). These features may attribute to the strong interaction between the solitons and background field.” in page 9 of the revised Supplementary Information and added new numerical simulations as Figs. S3e, f on the chaotic states.

Also, we did not generate soliton crystals or Turing rolls in the experiments, which may be attributed to the limited frequency range or the red detuning (the generation of the Turing rolls needs blue-detuned pump) of the Brillouin laser, since the frequency variation of the generated Brillouin laser is much smaller than the pump laser (*Phys. Rev. Lett.* **126**, 063901 (2021)). To reveal the underlying physics, we will take more investigations in the future. To address this question, we have added a sentence: “In the experiment, we did not observe the soliton crystals (or Turing rolls), which may be attributed to the limited frequency range (or the red detuning) of the generated Brillouin laser. This is because the frequency tuning range of the generated

Brillouin laser is much smaller than the input pump¹.” in page 9 of the revised Supplementary Information.

In Reviewer 1’s Report: The reference list lacks important papers related to the claims reported in this contribution. They should include those references: [1-7].

Our Response: We are grateful to the reviewer for pointing out these interesting references to us. We have included these important references in the revised manuscript as Refs. [38, 39, 42, 43, 47, 49, 50].

Response to Reviewer 2:

In Reviewer 2’s Report: The authors describe the unique dynamics of the Brillouin Kerr microcomb generation experimentally and numerically, whose first demonstration is presented in their previous work (Y. Bai, et al., PRL 126, 063901 (2021)). The authors discuss the transition processes of the comb and the physics behind them from the viewpoints such as the gain competition and phase diagram. One of the critical features of the comb is the reliable (consistently reproducible) turnkey initiation of a single soliton state by simply adjusting the pump laser current, which is experimentally confirmed. In addition, the authors perform the microwave generation at K-band (23 GHz), and low levels of phase noise and frequency instability are achieved thanks to the laser noise suppression via the Brillouin scattering and the self-thermal stability of the comb, whose results are consistent with the demonstration in their previous study.

Our Response: We sincerely thank the reviewer for his/her appreciation of our current work.

In Reviewer 2’s Report: Although the described physics is interesting and the comb’s nature is intriguing for applications, I think the current edition of the work is not significant enough to be published in Nature Communications mainly due to two reasons: The comb and its effects, including low noise microwave generation, have been reported in their published work, and the described explanation about the physics in this study lacks some clarification. The followings are specific concerns, questions, and suggestions.

Our Response: We thank the reviewer for the helpful comment. We agree with the reviewer that the Brillouin-Kerr soliton microcomb was reported in our previous work (*Phys. Rev. Lett.* **126**, 063901 (2021)). However, that work exhibited multi-stability and no strong interaction between the CW background and solitons was observed. In the revised manuscript, we have made substantial changes both in the presentation and in addressing the technical issues raised by the reviewer. In particular, we have provided more physical explanations to the strong interaction between the CW background and solitons. We have also performed several new measurements such as generating the monostable single soliton with much higher pump power, characterizing the detuning of the generated Brillouin laser, optimizing and analyzing the phase noise of the generated microwave signal with turnkey operation, measuring the dispersion of pump and Brillouin mode families as well as the powers and linewidths of Brillouin laser with different input pump laser. We sincerely hope that the new experimental results and significantly revised manuscript could be satisfactory to the reviewer.

In Reviewer 2's Report: The red-detuning of the Brillouin light can be a more critical factor for directly accessing the soliton states than the interaction between background wave and combs. Discussion about blue-detuned Brillouin pumping should be provided to deepen the understanding of physics rather than keeping the red-detuned condition untouched prerequisite.

Our Response: We thank the reviewer for this helpful suggestion. In our previous work (*Phys. Rev. Lett.* **126**, 063901 (2021)), the red-detuned Brillouin laser with a blue-detuned pump was demonstrated using theoretical simulations. Here, to deepen the understanding of physics, we have measured the detuning of the generated Brillouin laser by launching a probe light while generating the soliton microcombs (*Nat. Commun.* **6**, 5668 (2015)). According to our measurement (Supplementary Fig. S10), we can further conclude that the generated Brillouin laser is indeed red-detuned. To address this import issue, we have followed the reviewer's suggestion and added the measured results as Section X in page 13 of the Supplementary Information.

In Reviewer 2's Report: Note that the intensity of CW background light is not fixed even in conventional numerical frameworks (as mentioned in supplementary material around line 113), where the external power flow is usually taken into account via detuning (phase) terms. The interaction between background wave and comb exists, not depending on if you supply driving power directly or indirectly.

Our Response: We sincerely thank the reviewer for his/her correct understanding of the CW background light and we apologize for our misrepresentation in the original Supplementary Information. We have correspondingly corrected this mistake in the revised manuscript. For convenience, we copy the revised sentences here: to "According to the theoretical framework of dissipative Kerr-soliton formation in microcavity^{3,4}, the solution for the CW background light, as a fixed quantity, originates from the detuning term $\Delta\omega'_p$ and driving term defined by the external pump laser, $F'_{driv} = \sqrt{\kappa'_+} s'_{in}$, where κ'_+ is the external coupling rate of the input pump field with the amplitude s'_{in} . Thereby, if the pump power, frequency detuning and the coupling condition of the cavity modes are all fixed, $\Delta\omega'_p$ and F'_{driv} will remain unchanged. Different from this conventional picture, in our Brillouin-Kerr soliton generation the detuning term $\Delta\omega_{b,0}$ and driving term F_{driv} in Eq. (S6) is variable instead fixed." in page 4 of the revised Supplementary Information.

In Reviewer 2's Report: The use of the word "degeneracy" in "the degeneracy of different soliton states" is confusing because each state has a different energy level, as the authors mention in the article (i.e., multi-stability and degeneracy should be different). Note that I agree that selectively accessing soliton states is difficult in standard microcomb generation, where we need to go through chaotic states.

Our Response: We thank the reviewer for pointing out this terminology issue. We have removed the term "degeneracy" throughout the whole text and replaced it with "multi-stability" as suggested.

In Reviewer 2's Report: Can you comment on the effects and requirements of resonator dispersion?

Our Response: We thank the reviewer for bringing up this question. Actually, we have discussed the requirements of resonator dispersions in our previous work (*Phys. Rev. Lett.* **126**, 063901 (2021)). In our system, the soliton microcombs are generated only in the mode family of the Brillouin mode. So, the mode family of the Brillouin mode should exhibit anomalous group velocity dispersion while the mode family of the pump mode does not need to satisfy this requirement. Also, we have measured the dispersions of both the mode families of the pump and the Brillouin modes. As shown in Fig. S8, the Brillouin mode family indeed exhibits an anomalous group velocity dispersion with a second-order dispersion parameter $D_2/2\pi$ of approximately 23.7 kHz. However, in our experiment, we did not observe Kerr comb in the mode family of the pump mode due to the strongly distorted dispersion by the avoided mode crossing (Fig. S8) and the relatively low Q-factor of the pump mode. To address this comment, we have added the measured dispersions as Fig. S8 and a new Section VIII in page 12 of the revised Supplementary Information.

In Reviewer 2's Report: Can you indicate what power level in Fig. 3e corresponds to the experimental condition for each trace of Fig. 3f, approximately?

Our Response: We thank the reviewer for this helpful comment. Here in the revised manuscript, we have replaced the arbitrary units with the absolute power values in Fig. 3e.

In Reviewer 2's Report: Do you experimentally observe breathing single soliton states (SS) obtained in your numerical simulation (Fig. 3e), though the corresponding state transition is not evident in the time evolution of transmittance?

Our Response: We thank the reviewer for carefully viewing our phase diagram of Fig. 3e. During our experiments, we did not observe the breathing single-soliton state, which might be attributed to the thermal instability of this state. Also, in our original simulations, there were two state regions (including the coexistence of single soliton and flat states as well as the coexistence of breathing single soliton and flat states) in the “(breathing) single soliton (SS)” area. To make our phase diagram more accurate, we have distinguished these two regions in the revised Fig. 3e. According to the revised simulation, we can see that the breathing single soliton only exists in a narrow parametric region below the OPO threshold.

In Reviewer 2's Report: Please add an explanation how you couple light into the resonator (e.g., tapered fiber, integrated waveguide).

Our Response: We thank the reviewer for this suggestion. In our experiments, we use the tapered fiber to couple light into the microtoroid resonator. To address this comment, we have added a sentence: “Experimentally, we use a tapered fiber to evanescently couple light into the optical microresonator⁵³.” in the “Methods” section of the revised manuscript. Accordingly, a new reference (*Phys. Rev. Lett.* **85**, 74 (2000)) on the fiber taper coupling to the microcavity has been added as Ref. 53 in the revised manuscript.

In Reviewer 2's Report: What is the origin of bumps on frequency and phase noise curves at around 2 MHz offset? Can you show the sensitivity limit of the measurement?

Our Response: We are grateful to the reviewer for bringing up this important question. The presence of the bump on the frequency noise spectrum of the Brillouin laser can be attributed to the coupling between the amplitude and phase in the microresonator, (*Phys. Rev. A* **91**, 053843 (2015); *Optica* **2**, 225 (2015)). The resonant frequency of the amplitude fluctuation of Brillouin laser is associated with the linewidth of the optical modes, which is estimated at ~ 2 MHz (dominated by the lifetime of pump mode) in our system (*Phys. Rev. A* **91**, 053843 (2015); *Optica* **2**, 225 (2015)). Furthermore, the bump on the phase noise of the generated microwave signal is transduced from the frequency noise of the Brillouin laser.

In our previous measurement, there is a gap between the tapered fiber and the microtoroid resonator, which makes the Brillouin laser to exhibit relatively large noise fluctuation. To improve such coupling stability and reduce the bumps on noise curves, we have touched the fiber to the microcavity when performing the new low-noise performance measurements of the turnkey single-soliton microcomb. After that operation, the “bumps” become much smaller than that in our previously results. To address this issue, we have added a sentence: “It is important to highlight that during the measurement of the low-noise microwave signal, the tapered fiber is brought into contact with the microtoroid resonator to enhance system stability.” in page 10 of the revised manuscript. Also, we have followed the reviewer’s suggestion and shown the sensitivity limit in revised Fig. 5c.

Response to Reviewer 3:

In Reviewer 3’s Report: This manuscript, “Strong Interaction between Solitons and Background Light in Brillouin-Kerr Microcombs,” presents both experimental and numerical investigations of novel phenomena in Kerr soliton microcombs, specifically within the unique Brillouin-Kerr microcomb system. Unlike conventional microcomb setups, the driving source for soliton microcombs in this system relies on intracavity Brillouin light propagating in the backward direction, which offers several advantages over an externally provided forward pump.

Our Response: We sincerely thank the reviewer for his/her deep appreciation our work and the positive comments.

In Reviewer 3’s Report: Although I acknowledge that the results presented are valuable and relevant to the microcomb research community, I have some concerns regarding the manuscript’s overall scientific impact, particularly in the context of Nature Communications. I have listed specific questions and comments below.

Our Response: We thank the reviewer for the helpful comment. In our revised manuscript, we have performed more important measurements to reveal the underlying mechanism and further clarified the unique physics of our system. We expect that our new manuscript would dispel the doubts of the reviewer on our scientific impact.

In Reviewer 3’s Report: From what I understand, the deterministic emergence of a single-soliton state from a flat state (only Brillouin background) is a key result of this paper. However, I believe that the underlying principle is not sufficiently explained in the current manuscript. Given that

intracavity energy is consumed and subsequently reduced to form solitons through OPO processes, it is only natural for the reduction of the background field (i.e., pump power) to affect the potential number of solitons. This interaction does not seem to be a unique feature of the Brillouin-Kerr system, and, unfortunately, I cannot identify significant novelty in this finding.

Our Response: We sincerely thank the reviewer for bringing up this important issue. To further explain the strong interaction between the generation solitons and the background light, we have modified the sentence: “As a consequence, in an ultra-high-quality (ultra-high-Q) microcavity, ... thereby indicating a strong interaction between the background field and generated Kerr solitons.” to “As a consequence, in an ultra-high-quality (ultra-high-Q) microcavity, the generated Kerr-solitons will greatly influence the power of Brillouin laser through optical parametric frequency conversion (Supplementary Section I) and strongly attenuate the intracavity energy of the background light (Fig. 1a). This attenuation of background light subsequently leads to a significant reduction in the parametric gain that is used to sustain the Kerr solitons. Here, we refer the strong interaction between the generated Kerr solitons and the background light to the situation that the intracavity energy of the background light is decreased from above the optical-parametric-oscillation (OPO) threshold to below it during the Kerr soliton generation process.” in page 4 of the revised manuscript.

Furthermore, concerning on the reduction of the background field, we are also aware of a similar comment from Reviewer 1. In the conventional Kerr soliton combs, the consumption of the pump light does not mean the reduction of the background field. Actually, the generation of the solitons does not affect the amplitude of the background light (*Nat. Photon.* **4**, 471-476 (2010); *Nat. Photon.* **8**, 145-152 (2014); *Phys. Rev. E* **97**, 042204 (2018)) with fixed input pump power and detuning, although the intracavity energy of the pump is decreased. As a specific example, in an efficient Kerr soliton microcomb (*Nat. Commun.* **13**, 1292 (2022)), 55% pump power can be converted to the Kerr comb. However, during that process, the reduction of the background light was still not observed (this can be learned from Figs. S8 (b) and (d) in the Supplementary Information of *Nat. Commun.* **13**, 1292 (2022)), since the solitons only locally influenced the CW background. In contrast, in our system the amplitude of the background field is associated with the generated solitons (even if the input pump power and detuning is fixed) due to the strong interaction between the background light and the generated solitons, which finally enables the monostable single-soliton generation. To make our statement more clear, we have added a sentence: “Note that this global damping behavior of the intracavity energy of the CW background field during the soliton formations cannot occur in conventional^{1,4,28} or even efficient Kerr soliton microcombs³⁸ with fixed pump detuning and power, where the solitons only locally interact with the background light.” in page 6 of the revised manuscript. Accordingly, a new reference on the CW background field has been added as Ref. 38 in the revised manuscript.

In Reviewer 3's Report: *In relation to the previous comment, do multi-soliton states initially emerge instead of a single-soliton state when a stronger pump power is provided (e.g., >100 mW), as reported in the authors' previous paper (PRL, 126, 063901 (2021))?*

Our Response: We are grateful for the reviewer's comment. Following the reviewer's suggestion, we performed experimental measurement at much higher pump powers. As shown in Fig. S9 of the revised Supplementary Information, we found that only the single-soliton state can be initially generated even as the pump power increased to 100 mW, which is different from our previous

work (*Phys. Rev. Lett.* **126**, 063901 (2021)). This measurement further confirmed the monostability of our system induced by the strong interaction of the background light and generated Kerr solitons. To address this comment, we add a new Section IX in our revised Supplementary Information.

In Reviewer 3's Report: On page 5 (paragraph 5), the authors describe that higher Q-factors lead to a more pronounced effective Brillouin nonlinearity, thereby enhancing the interaction strength. What is meant by "effective Brillouin nonlinearity" in this context?

Our Response: We thank the reviewer for this helpful comment. The "effective Brillouin nonlinearity" refers to the effective Brillouin gain, which is induced by the stimulated Brillouin scattering. To make it clear, we have revised the term "effective Brillouin nonlinearity" to "effective Brillouin gain" in page 5 of the revised manuscript.

In Reviewer 3's Report: Although high-Q factors do lower the threshold power of Brillouin lasing, it is unclear how this effect directly relates to the interaction strength between Kerr combs and the Brillouin background field.

Our Response: We thank the reviewer for bringing up this question which allows us to further elaborate the advantages of our work. Actually, we have theoretically explained this in the Supplementary Information (page 5). To address this important issue, we have modified the sentence: "In this way, the interaction strength between the Brillouin laser and the generated Kerr-soliton microcomb will become more pronounced." to "In this way, the interaction strength between the Brillouin laser and the generated Kerr-soliton microcomb will become more pronounced, since the generation of Kerr solitons will remarkably reduce the intracavity energy of the Brillouin wave (see detailed discussions in the Supplementary Section I)" in page 5 of the revised main text.

In Reviewer 3's Report: It would be beneficial to include a comparison table of phase-noise performances from previous studies (particularly in the same material platforms).

Our Response: We thank the reviewer for the precious advice. We are also aware of a similar comment from Reviewer 1. We have followed the suggestion of the reviewer and added a table in the Supplementary Information XI to compare the phase noise of the generated microwave signal to previous studies.

In Reviewer 3's Report: I recommend adding the specific term "Brillouin-Kerr" to the title.

Our Response: We thank the reviewer for this sincere suggestion. Following the reviewer's suggestion, we have revised the title to "Strong interactions between solitons and background light in Brillouin-Kerr microcombs".

In Reviewer 3's Report: The Methods section could be expanded upon. For example, the fabrication methods of resonators and experimental details of soliton generation and measurements could be described in greater detail.

Our Response: We appreciate the reviewer's suggestion. We have added the "Methods" section to provided more details in revised manuscript, such as the fabrication and coupling of resonators, as well as measurement of phase noise of turnkey single soliton.

Reviewer #1 (Remarks to the Author):

Review of 'Strong interactions between solitons and background light in Kerr microcombs' by Menghua Zhang, Shulin Ding, Xinxin Li, Keren Pu, Shujian Lei, Min Xiao, and Xiaoshun Jiang.

The paper demonstrates an elegant new method to directly generate single solitons in a silica micro-resonator without exciting other states, as is the norm with conventional pumping. In the revised manuscript, the authors performed many modifications to the first version of the paper that, in my view, greatly improved its overall quality. All my concerns and doubts have been addressed regarding the scientific content and some of the terminology that the authors employed in the first version.

Therefore, I strongly recommend the publication of this paper in Nature Communications.

Reviewer #2 (Remarks to the Author):

I appreciate the authors' effort to give answers to the questions and improve the quality of the paper. I acknowledge that some elaboration and clarification have been made. However, the paper still lacks a reasonable explanation of the authors' central claim and novelty to meet Nature Communication's high standards demands. The following are my specific concerns.

I do not think the authors provide sufficient clues and explanation of the physics behind the "strong" interaction and induced significant background power drop, which are the critical claims of this study.

-First, can you elaborate on the difference between "global" and "local" interaction between the CW background and solitons? Regarding the spatial overlap between background and solitons, there should be no difference between the conventional case and this study. It is unclear what the words are defined for.

-Though the authors mention the power drop has not been observed in previous studies by citing a reference (i.e., "this can be learned from Figs. S8 (b) and (d) in the Supplementary Information of Nat. Commun. 13, 1292 (2022)"), I do not think the figures clarify the point because the power ratios between the pulse peak and background in the rings are not so different in the two cases (blue curve in Fig.S8(d) in the ref gives ~17%. The paper shows ~13 % in Fig.3(b). It is natural that the output and feedback ports give a higher background since it contains a CW pump light that does not couple to the resonator). It can also depend on various simulation parameters. To see the effect, the temporal evolution of comb states needs to be compared fairly, though I believe the background power drop is not unusual in conventional soliton generation processes (e.g., Fig.2D of Science 361,eaan8083(2018)).

-Although I suppose the equation S21 explains the physics behind the power dump in the authors' theoretical framework, what is the reason why it does not include the remaining pump term that is not converted into the Brillouin wave but still exists inside the resonator?

-I feel the points the authors answered with the sentence "may be attributed to" (e.g., "Such chaotic motions may be attributed to the interaction between the solitons and background field.") are closely related to the core claim and their underlying physics needs to be revealed before the publication.

-As to my question about detuning, I am sorry if the point is unclear. I thought the red-detuned Brillouin pump could allow direct access to a single soliton state (in the red detuning side in the phase diagram), thanks to the blue-detuned pump that maintains thermal stability. This time, I would like to put a similar question differently. Do you think it is possible to generate a soliton in a similar way to this study by using two independent lasers, where one is at the blue side of a resonance, and the other is at the red side of another resonance? The latter behaves the same way as the Brillouin pump

in this study regarding power and detuning, but there is no nonlinear coupling between them.

-I find the word usage, such as "To break the multi-stability between single- and multi-soliton states" and "we find that the multi-stability between single-soliton state and flat or multi-soliton states will break down.", is strange because they are separate states. Although I supposed this paper claims the selective and reproducible access to a state (single soliton) or prohibiting from accessing other states, do you want to suggest they are originally "degenerated" and the demonstrated pumping scheme breaks them down?

Response to Reviewer 1:

In Reviewer 1's Report: The paper demonstrates an elegant new method to directly generate single solitons in a silica micro-resonator without exciting other states, as is the norm with conventional pumping. In the revised manuscript, the authors performed many modifications to the first version of the paper that, in my view, greatly improved its overall quality. All my concerns and doubts have been addressed regarding the scientific content and some of the terminology that the authors employed in the first version. Therefore, I strongly recommend the publication of this paper in Nature Communications.

Our Response: We sincerely thank the reviewer for his/her positive comments on our revised manuscript and for the recommendation of publishing it on Nature Communications.

Response to Reviewer 2:

In Reviewer 2's Report: I appreciate the authors' effort to give answers to the questions and improve the quality of the paper. I acknowledge that some elaboration and clarification have been made. However, the paper still lacks a reasonable explanation of the authors' central claim and novelty to meet Nature Communication's high standards demands. The following are my specific concerns.

Our Response: We sincerely appreciate the reviewer for acknowledging our efforts on the previous responses. To address the reviewers' comments, we have thoroughly explained and revised the manuscript so that it meets the highly standard demands of Nature Communications.

In Reviewer 2's Report: I do not think the authors provide sufficient clues and explanation of the physics behind the "strong" interaction and induced significant background power drop, which are the critical claims of this study.

Our Response: Regarding this comment of the reviewers, we have thoroughly revised the manuscript again so that the reviewer's concern can be clarified.

In addition to the revision made in the manuscript, we summarize the relevant physics as follows:

1. To provide more clues of such strong interaction, we have numerically calculated the emergence of the monostable single soliton as a function of the loaded Q-factor of the Brillouin optical mode, under the condition of unchanged

system parameters except for the intrinsic decay rate of the Brillouin optical mode (please refer to the calculated results in Fig. R1). It shows that the monostable single soliton will appear given that the loaded Q-factor is larger than 3.3×10^8 (the corresponding experimental value is 4.4×10^8), which further demonstrates that the strong interaction is due to the ultra-high Q-factor of the optical microresonator in the Brillouin-Kerr soliton system. To address this point, on page 5 of the revised manuscript, we have changed the sentence: “In this way, the interaction strength between the Brillouin laser and the generated Kerr-soliton microcomb will become more pronounced, since the generation of Kerr solitons will remarkably reduce the intracavity energy of the Brillouin wave (see detailed discussions in the Supplementary Section I).” to “In this way, the interaction strength between the Brillouin laser and the generated Kerr-soliton microcomb will become more pronounced (Supplementary Fig. S6), since the generation of Kerr solitons will remarkably reduce the intracavity energy of the Brillouin wave (see detailed discussions in the Supplementary Section I)”. We have added a new Section “V. The emergence of the monostable single soliton determined by Q-factor” on page 11 of the revised Supplementary Information. Moreover, we added the following Fig. R1 to the revised Supplementary Information as Fig. S6.

Fig. R1 The existence diagram of the monostable single-soliton states as a function of input pump power and the loaded Q-factor of the Brillouin optical mode. Dashed red curve plots the OPO threshold of the Brillouin laser under the optimal pump detuning.

2. For a better description of the strong interaction between the generated solitons and background field, we have modified the sentence: “Here, we refer the strong interaction between the generated Kerr solitons and the background light to the situation that the intracavity energy of the background light is decreased from above the optical-parametric-oscillation (OPO) threshold to below it during the Kerr soliton generation process.” to “Here, the strong interaction between the generated Kerr solitons and the background light is referred to the situation, in which the

intracavity energy of the background light is decreased from above to below the optical-parametric-oscillation (OPO) threshold and becomes low enough to prevent the formation of multi-solitons during the Kerr soliton generation process.” on page 5 of the revised manuscript.

In Reviewer 2's Report: First, can you elaborate on the difference between “global” and “local” interaction between the CW background and solitons? Regarding the spatial overlap between background and solitons, there should be no difference between the conventional case and this study. It is unclear what the words are defined for.

Our Response: In conventional Kerr soliton combs, the propagation loss of solitons is compensated by the nonlinear interaction between the solitons and background in their spatially overlapped region (*Opt. Express* **21**, 28862 (2013); *Nat. Photon* **8**, 145 (2014); *Nat. Photon* **4**, 471 (2010)). Despite the presence of this local and weak interaction, the CW background is unaffected by the generation of Kerr solitons (*Opt. Express* **21**, 28862 (2013); *Nat. Photon* **8**, 145 (2014); *Nat. Photon* **4**, 471 (2010)), as shown in Fig. R2a. However, as shown in Fig. R2b, in the process of our Brillouin-Kerr soliton generation, the intracavity energy of background light will globally reduce at every position inside the resonator (even outside the overlap region) due to the strong interaction mechanism. This phenomenon, which we refer to as global reduction, is a distinctive characteristic of our Brillouin-Kerr soliton system. Certainly, we agree with the Reviewer that the behavior of spatial overlap between the background and generated solitons in our Brillouin-Kerr soliton has no difference from those in a conventional case.

To clarify the notion “localized interaction” further, we have modified the sentence: “However, in previous Kerr soliton combs, the intrinsic interactions between the background light and solitons are typically weak and localized^{4,5}, which means that the formation of solitons has little impact on the amplitude of the background light.” to “However, in previous Kerr soliton combs, the intrinsic interactions between the background light and solitons are typically weak and localized in the spatial overlap region^{4,5,26}, which means that the formation of solitons has little impact on the amplitude of the background light.”. This revision is on page 3 of the revised manuscript.

On the other hand, to clarify the concept “global damping behavior” further, we have added a new Section “III. Comparison of the generation of the monostable single-soliton to the conventional Kerr solitons” on page 7 of the revised Supplementary Information. Moreover, on page 7 of the revised manuscript, we have modified the sentence: “Note that this global damping behavior of the intracavity energy of the CW background field during the soliton formations cannot occur in conventional^{1,4,28} or even efficient Kerr soliton microcombs³⁸ with fixed pump detuning and power, where the

solitons only locally interact with the background light.” to “Note that this global damping behavior (Supplementary Section III) of the intracavity energy of the CW background field during the soliton formations cannot occur in a conventional^{1,4,29} or even an efficient Kerr soliton microcombs³⁹ under the fixed pump detuning and power, where the solitons only locally interact with the background light.”. We have also added the following Fig. R2 to the revised Supplementary Information as Fig. S1.

Fig. R2 Numerical simulations on the single-soliton states. **a**, blue line, single soliton simulated by LLE model; red line, corresponding flat state before the soliton generation. **b**, blue line, monostable single soliton excited in Brillouin-Kerr frequency comb system; red line, corresponding flat state before the soliton generation.

In Reviewer 2's Report: Though the authors mention the power drop has not been observed in previous studies by citing a reference (i.e., “this can be learned from Figs. S8 (b) and (d) in the Supplementary Information of *Nat. Commun.* 13, 1292 (2022)”), I do not think the figures clarify the point because the power ratios between the pulse peak and background in the rings are not so different in the two cases (blue curve in Fig.S8(d) in the ref gives ~17%. The paper shows ~13 % in Fig.3(b). It is natural that the output and feedback ports give a higher background since it contains a CW pump light that does not couple to the resonator). It can also depend on various simulation parameters. To see the effect, the temporal evolution of comb states needs to be compared fairly, though I believe the background power drop is not unusual in conventional soliton generation processes (e.g., Fig.2D of *Science* 361, ean8083 (2018)).

Our Response: In fact, we have cited the references of the typical soliton generation processes such as *Science* 361, ean8083 (2018); *Nat. Photon* 4, 471 (2010); *Phys. Rev. E* 97, 042204 (2018). We would like to highlight that even in the high-efficiency soliton generation (*Nat. Commun.* 13, 1292 (2022)), the backgrounds in the cavity of multiple solitons and single soliton are equal under the same parameters as shown in Figs. S8 (b) and (d) in Supplementary Information of *Nat. Commun.* 13, 1292 (2022). It is obvious that, under the high efficiency soliton condition, there does not exist a

global power drop of background among different soliton states.

Following the advice of the Reviewer, we have numerically simulated the generations of the single solitons both in the conventional Kerr soliton and in our Brillouin-Kerr soliton for the sake of a fair comparison.

In the conventional Kerr soliton formation process, the intracavity energy of the CW background field for the soliton state is equal to that of the flat state under the same conditions of the same pump power, the same pump detuning, and the same external coupling rates, as shown in Fig. R2a. However, in our Brillouin-Kerr soliton generation (Fig. R2b), a noticeable difference is that the intracavity energy of the background light will decrease after the Brillouin-Kerr soliton is generated. To address this issue, we have added a new Section “III. Comparison of the generation of the monostable single-soliton to the conventional Kerr solitons” to the revised Supplementary Information.

In Reviewer 2's Report: Although I suppose the equation S21 explains the physics behind the power dump in the authors' theoretical framework, what is the reason why it does not include the remaining pump term that is not converted into the Brillouin wave but still exists inside the resonator?

Our Response: Actually, in our Brillouin-Kerr soliton microcombs, the CW background directly comes from the intracavity Brillouin laser related to the input pump. Therefore, we establish the direct relationship (Eq. S21) between the amplitudes of the CW background field A_{back} and the generated Brillouin wave a_- . $F_{driv} = -ig_b a_+ b^*$ (in Eq. S6) is the driving term for the Brillouin laser generation, where a_+ is the amplitude of the remaining pump field after the Brillouin-Kerr soliton generations. After we substitute Eqs. S10 and S13 into this driving term, the two variables of a_+ and b^* will be eliminated. Then, we obtain Eq. 21 about the two fields A_{back} and a_- .

In Reviewer 2's Report: I feel the points the authors answered with the sentence “may be attributed to” (e.g., “Such chaotic motions may be attributed to the interaction between the solitons and background field.”) are closely related to the core claim and their underlying physics needs to be revealed before the publication.

Our Response: We thank the reviewer for bringing up this helpful comment. Following the suggestion of the reviewer, we show more simulation details (see Fig. S5g of the revised Supplementary Information) of the chaotic states. From these simulations, we can find that the number of solitons and the energy of the background light exhibit variations over the time. This observation is related to the interaction between the solitons and the background field in our Brillouin-Kerr soliton system. To make our statement clearer, we have modified the sentence: “These features may attribute to the strong interaction between the solitons and background field.” to “This observation

is related to the interaction between the solitons and the background field in our Brillouin-Kerr soliton system. The detailed study on such chaotic motions is beyond the scope of this work, which will be further studied in the future.” on page 11 of the revised Supplementary Information.

In Reviewer 2's Report: As to my question about detuning, I am sorry if the point is unclear. I thought the red-detuned Brillouin pump could allow direct access to a single soliton state (in the red detuning side in the phase diagram), thanks to the blue-detuned pump that maintains thermal stability. This time, I would like to put a similar question differently. Do you think it is possible to generate a soliton in a similar way to this study by using two independent lasers, where one is at the blue side of a resonance, and the other is at the red side of another resonance? The latter behaves the same way as the Brillouin pump in this study regarding power and detuning, but there is no nonlinear coupling between them.

Our Response: We thank the reviewer for raising this important comment. Indeed, the method described by the reviewer can generate Kerr solitons, which has been well demonstrated in a previous study (*Optica* **6**, 206 (2019)). Although the independent auxiliary laser solves the issue of thermally stable entrance of the soliton states, the soliton existence range for different states is still degeneracy, which is owing to the pump laser can never be affected by the generation of Kerr solitons.

To experimentally compare our scheme with that method, we directly pump the Brillouin mode at 1561.4 nm to generate the conventional Kerr solitons by using an independent auxiliary laser at 1557.3 nm to suppress the thermo-optic effect. As shown in Fig. R3a, discrete and stochastic soliton steps are obviously observed in the transmission. This phenomenon indicates that the soliton existence range for different states is still in degeneracy. It is worth mentioning that the spectra generated here (Fig. R3b) are slightly different from Fig. 2b in the main text due to the increased coupling between the fiber taper and the microcavity.

To address this comment, we have added the following Fig. R3 to the revised Supplementary Information as Fig. S2. Moreover, we have modified the sentence: “Additionally, we find that the single-soliton state can be 100% deterministically generated (by repeatedly sweeping the pump laser frequency, see Supplementary Fig. S4).” to “Additionally, we find that the single-soliton state can be 100% deterministically generated (by repeatedly sweeping the pump laser frequency, see Supplementary Fig. S7), which is quite different from the generation of the conventional Kerr solitons by directly pumping the Brillouin mode (Supplementary Fig. S2).” on page 7 of the revised manuscript.

Fig. R3 a, The measured transmission of pump laser at 1561.4 nm and auxiliary laser at 1557.3nm. The steps existing in the red-detuning regimes of the Brillouin mode. **b**, The measured optical spectra of solitons corresponding to (a).

In Reviewer 2's Report: I find the word usage, such as "To break the multi-stability between single- and multi-soliton states" and "we find that the multi-stability between single-soliton state and flat or multi-soliton states will break down.", is strange because they are separate states. Although I supposed this paper claims the selective and reproducible access to a state (single soliton) or prohibiting from accessing other states, do you want to suggest they are originally "degenerated" and the demonstrated pumping scheme breaks them down?

Our Response: We thank the reviewer for this very helpful comment. Here, we refer to "multi-stability" as the existence range of multiple stable soliton states given the same pump detuning, which leads to the degeneracy of the soliton existence range for the different states in conventional Kerr soliton system. To well address this issue, we have changed the sentence: "As a result, solitons with different numbers usually share the same CW background field under the same pump condition^{3,26-28}, which leads to the multi-stability of different soliton states." to "As a result, solitons with different numbers usually share the same CW background field under the same pump condition^{5,27-29}, which leads to the degeneracy of the soliton existence range for different states, i.e. the multiple stable soliton states exist under the same pump detuning."

We also revised two other sentences: "To break the multi-stability between single- and multi-soliton states..." and "we find that the multi-stability between single-soliton state and flat or multi-soliton states will break down." to "To break the degeneracy..." and "we find that the degeneracy of the existence range between the single soliton state and the flat or multi-soliton states", respectively.

Reviewer #2 (Remarks to the Author):

I appreciate the authors for demonstrating detailed additional experiments and numerical calculations. All concerns have been addressed, and the paper has become clearer and more informative with the revisions. I am convinced that the observed effect is distinctive from the previous reports. I now support its publication in Nature Communications.

Correspondence (NCOMMS-23-12889-B) to Reviewers' Reports

Response to Reviewer 2:

In Reviewer 2's Report: I appreciate the authors for demonstrating detailed additional experiments and numerical calculations. All concerns have been addressed, and the paper has become clearer and more informative with the revisions. I am convinced that the observed effect is distinctive from the previous reports. I now support its publication in Nature Communications.

Our Response: We sincerely thank the reviewer for the positive comments and recommendation for publication on our work.